# A Novel Freshwater Cyanophage, Mae-Yong924-1, Reveals a New Family

**DOI:** 10.3390/v14020283

**Published:** 2022-01-28

**Authors:** Minhua Qian, Dengfeng Li, Wei Lin, Lingting Pan, Wencai Liu, Qin Zhou, Ruqian Cai, Fei Wang, Junquan Zhu, Yigang Tong

**Affiliations:** 1School of Marine Sciences, Ningbo University, Ningbo 315211, China; qianminhua201@163.com (M.Q.); weilin0577@163.com (W.L.); 1911091090@nbu.edu.cn (L.P.); liuwencai103@163.com (W.L.); zhouqin0987@163.com (Q.Z.); cairuqian824@163.com (R.C.); a978915665@163.com (F.W.); zhujunquan@nbu.edu.cn (J.Z.); 2College of Life Science and Technology, Beijing University of Chemical Technology, Beijing 100029, China

**Keywords:** complete genome, freshwater phage, *Microcystis virus*, *Microcystis aeruginosa*, new clade

## Abstract

Cyanobacterial blooms are a worldwide ecological issue. Cyanophages are aquatic viruses specifically infecting cyanobacteria. Little is known about freshwater cyanophages. In this study, a freshwater cyanophage, Mae-Yong924-1, was isolated by the double-layer agar plate method using *Microcystis aeruginosa* FACHB-924 as an indicator host. Mae-Yong924-1 has several unusual characteristics: a unique shape, cross-taxonomic order infectivity and a very unique genome sequence. Mae-Yong924-1 contains a nearly spherical head of about 100 nm in diameter. The tail or tail-like structure (approximately 40 nm in length) is like the tassel of a round Chinese lantern. It could lyse six diverse cyanobacteria strains across three orders including *Chroococcales*, *Nostocales* and *Oscillatoriales*. The genome of the cyanophage is 40,325 bp in length, with a G + C content of 48.32%, and 59 predicted open reading frames (ORFs), only 12 (20%) of which were functionally annotated. Both BLASTn and BLASTx scanning resulted in “No significant similarity found”, i.e., the Mae-Yong924-1 genome shared extremely low homology with sequences in NCBI databases. Mae-Yong924-1 formed a root node alone and monopolized a root branch in the proteomic tree based on genome-wide sequence similarities. The results suggest that Mae-Yong924-1 may reveal a new unknown family apparently distinct from other viruses.

## 1. Introduction

Blooms of toxic cyanobacteria have become a common occurrence in water bodies worldwide. The frequency and intensity of cyanobacterial blooms are becoming increasingly serious. Cyanobacterial blooms consume a large amount of dissolved oxygen in the water and release cyanotoxins, endangering aquaculture objects, resulting in food safety risks to products due to algal toxin residues and polluting water sources [1]. Cyanotoxins are key risk factors that contribute to the development of liver cancer. The high incidence of primary liver carcinoma (PLC) was found to correlate with emerging blooms of cyanobacteria in water reservoirs supplying drinking water [2]. *Microcystis* spp. are the most ecologically harmful and dominant bloom-forming cyanobacteria in freshwater lakes [3]. They usually produce microcystins (MCs) at concentrations that cause chronic poisoning or acute death to animals and humans [4]. MCs have been intensively studied among cyanotoxins due to their toxicity, causing severe liver and kidney damage, tumor promotion and gastroenteritis [5,6]. Accumulation of MCs in aquatic ecosystems increases public health concerns [7,8].

Effective control of water blooms has become a top priority. Cyanophages, aquatic viruses that specifically infect cyanobacteria, are key factors that mediate host communities, the food web, carbon cycling and nutrient recycling. Cyanophages, playing a key role in regulating the population of cyanobacteria, can be used as an environmentally friendly biological agent to control cyanobacterial blooms [9,10]. Most of the studies on cyanophages were focused on marine cyanophages, especially cyanophages of *Prochlorococcus* and *Synechococcus* [11,12]. There are few reports on freshwater cyanophages and *Microcystis* phages. To date, only 10 *Microcystis* cyanophages have been reported [13], of which only 5 genomes have been sequenced and characterized. The full genome sequenced *Microcystis* cyanophages are MaMV-DC, Ma-LMM01, Mic1, vB_MelS-Me-ZS1 and MaAM05 (alias PhiMa05), harboring 169,223 bp, 162,109 bp, 92,627 bp, 49,665 bp and 273,876 bp [13,14,15,16,17] dsDNA molecules, respectively. *Microcystis aeruginosa* forms toxic cyanobacterial blooms throughout the world. Yet, thus far, only four reported cyanophages have been isolated with *M. aeruginosa*, which are Ma-LBP, Ma-LMM01, MaMV-DC and Mic1 [14,15,16,17]. Ma-LBP displays a *Podoviridae*-like morphology; Ma-LMM01, MaMV-DC and PhiMa05 display a *Myoviridae*-like morphology; and Mic1 and vB_MelS-Me-ZS1 display a *Siphoviridae*-like morphology. Ma-LMM01 and MaMV-DC have been classified as *Myoviridae* viruses by the ICTV.

Here, we isolated a novel freshwater cyanophage, Mae-Yong924-1, using *M. aeruginosa* FACHB-924 as an indicator cyanobacterial strain. Mae-Yong924-1 has several unusual characteristics: cross-taxonomic order infectivity, a special shape and a unique genome sequence, forming a root node alone and monopolizing along an independent root branch in the phylogenetic tree based on complete genome similarities. We propose that Mae-Yong924-1 may represent a new family distinct from the other viruses.

## 2. Materials and Methods

### 2.1. Isolation and Amplification of Cyanophage

Surface water samples were collected from the Yangming Lake (north latitude, 29.773222; east longitude, 121.954422) in the Meishan campus of Ningbo University. The water samples were centrifuged (10,000× *g*, 20 min) and filtered (through 0.45 µm and 0.22 µm nitrocellulose filter membranes (ANPEL Laboratory Technologies Inc, Shanghai, China)) as described [13]. The enrichment and separation of the cyanophage were carried out according to the process reported in [13], employing *M. aeruginosa* FACHB-924 as the host. In brief, 3 mL aliquots of the filtrates were mixed with FACHB-924 cultures at the exponential phase (at the volume ratio of 1:2) and cultured for 10–20 days until the cultures were etiolated. The yellowing lysate was centrifuged, filtered and applied to the next round of enrichment, as described above. The enrichment was performed for three rounds. A pure cyanophage strain was obtained by five serial single-plaque isolations using the double-layer agar method [13]. A single plaque was dug out from the plate, mixed with 3 mL of exponentially growing FACHB-924 and cultured in a light incubator for 10–15 days. The yellowed and clarified culture suspension was centrifuged (6000× *g*, 20 min) and filtered (0.45 µm and 0.22 µm). Amplification culture was performed via co-cultivation of *M. aeruginosa* FACHB-924 and the filtrate containing the *Microcystis* cyanophage Mae-Yong924-1 (at a volume ratio of 5:1) until the culture turned yellow (for about 10 days).

### 2.2. Electron Microscopy

Mae-Yong924-1 suspensions were dropped onto 400-mesh copper grids and negatively stained for 30 s with 2% uranyl acetate (Sigma-Aldrich, St. Louis, MO, USA). Photographs were taken with a TEM (Hitachi-7650, Tokyo, Japan) with a magnification of 80,000×.

### 2.3. Host Range Determination

All 14 tested cyanobacteria (Table 1) were obtained from the freshwater Algae Culture Bank, from the Institute of Hydrobiology (Wuhan, China) of the Academy of Sciences. Three hundred-microliter aliquots of the Mae-Yong924-1 suspension were added to the exponential cultures (600 µL) of the tested cyanobacterial strains in triplicate in 48-well cell culture plates and incubated in a light incubator. The phage suspension was replaced with BG11 medium in the negative control groups. All cultures were monitored daily for cell lysis via visual inspection and optical microscopy (VersaMax, Molecular Devices, Sunnyvale, CA, USA) observation and checked by measuring the absorbance at OD_680_. Cyanobacterial strains that did not lyse until the 12th day were defined as insusceptible.

### 2.4. Genome Extraction, Sequencing and Assembly

The cyanophage genome was extracted using a High Pure Viral RNA Kit (Roche, Basel, Switzerland, product no. 11858882001). This kit allows the extraction of DNA and RNA together. The NEBNext Ultra™ II DNA Library PrepKit for Illumina was used for constructing a 2 × 300 bp paired-end DNA library. The Illumina MiSeq (San Diego, CA, USA) sequencing platform was used for sequencing. FACHB-924 culture without the cyanophage was performed in the same way as for the control group. The reads appearing in the control group were deleted from the raw sequencing data of the experimental group. Fastp and Trimmomatic-0.36 were used to filter out bad reads (Q-value < 20) and adapters. SPAdes 3.13.0 (http://cab.spbu.ru/software/spades/; 16 February 2021) was used to assemble the clean reads. Phage genome termini were predicted by the proposed method in [18].

### 2.5. Genome Annotation

Open reading frames (ORFs) were initially predicted with the RAST annotation server (http://rast.nmpdr.org/; 17 February 2021) [19]. All predicted ORFs were verified by utilizing BLASTp (E-value ≤ 10^−5^), Hmmer (https://www.ebi.ac.uk/Tools/hmmer/search/hmmscan; 10 November 2021) (E-value ≤ 10^−5^) and HHpred (https://toolkit.tuebingen.mpg.de/#/tools/hhpred; 10 November 2021) [20] (E-value ≤ 10^−5^, percentage possibility of homologous sequences > 96%). The CRISPR spacers in the *Microcystis* cyanophage Mae-Yong924-1 genome were identified using the CRISPRs web server (https://crispr.i2bc.paris-saclay.fr/; 2 November 2021) [21,22] and Integrated Microbial Genome/Virus (IMG/VR) version 3 (https://img.jgi.doe.gov/; 2 November 2021) [23].

### 2.6. Phylogenetic Tree

The proteomic tree approach is effective in investigating genomes of newly sequenced viruses. ViPTree online [24,25] (https://www.genome.jp/viptree/; 10 December 2021) was used to generate a proteomic tree based on genome-wide sequence similarities, computed by tBLASTx.

## 3. Results and Discussion

After five successive single-plaque isolations, uniform plaques were obtained. Mae-Yong924-1 is the fifth cyanophage isolated with the cyanobacterium *M. aeruginosa*. The lysis of the co-cultures of FACHB-924 with the cyanophage Mae-Yong924-1 occurred in 7–10 days.

Observation under TEM found that negatively stained Mae-Yong924-1 possesses a nearly spherical head of approximately 100 nm in diameter. The tail or tail-like structure (approximately 40 nm in length) of Mae-Yong924-1 is unique, which is like the tassel of a round Chinese lantern (Figure 1).

In the host range experiments, lytic efficiencies occurred in 6 of the 14 tested cyanobacterial strains (Table 1). The susceptible cyanobacterial strains were *M. aeruginosa* FACHB-924, *M. elabens* FACHB-916, *Microcystis* sp. PCC-7806 of *Chroococcales*, *Nostoc* sp. FACHB-596, *Aphanizomenon flos-aquae* FACHB-1209 of *Nostocales* and *Planktothricoides raciborskii* FACHB-881 of *Oscillatoriales* (Table 1). The results demonstrate that the *Microcystis* cyanophage Mae-Yong924-1 had the ability to lyse cyanobacterial strains across taxonomic orders (*Chroococcales*, *Nostocales* and *Oscillatoriales*). A wider host range may be advantageous for the application because cyanobacterial blooms are usually caused by multiple cyanobacteria [9]. Although most isolated and well-studied phages have a narrow host range, an increasing number of broad-host-range phages have been found recently [26]. The *Aquamicrobium* phage P14 has even been found to infect various bacterial strains across taxonomic classes, *Alphaproteobacteria* and *Betaproteobacteria* [27]. Among the six susceptible cyanobacterial strains, the lysis of *M. elabens* FACHB-916 by Mae-Yong924-1 appeared to be the most efficient. Mae-Yong924-1 could fully lyse *M. elabens* FACHB-916 in the exponential phase in 1–3 days. Microscopic observation demonstrated that the number of intact cyanobacterial cells in the yellowing tested groups was significantly less than that in the control groups (Figure 2). Subsequently, the amplification of the cyanophage Mae-Yong924-1 was performed via co-cultivation of Mae-Yong924-1 with *M. elabens* FACHB-916 (at a volume ratio of 5:1). A literature search indicated that Mae-Yong924-1 is the second cyanophage found to be able to rapidly lyse *M. elabens* FACHB-916.

The complete genome sequence of Mae-Yong924-1 was sequenced with an average sequencing depth of 169-fold. The Mae-Yong924-1 genome is 40,325 bp in length with a G + C content of 48.32%. BLASTn and BLASTx analyses resulted in “No significant similarity found”, i.e., the results illustrated that the Mae-Yong924-1 genome shared extremely low homology with the sequences in current databases. Phage terminus analysis showed that Mae-Yong924-1 had no fixed phage terminus. The genome was deposited in GenBank under the accession number MZ447863.

The CRISPR/Cas immune system is a defense strategy against foreign nucleic acids such as phage genomes of bacteria and archaea [28]. CRISPR loci are normally composed of discontinuous direct repeats separated by short stretches of DNA sequences called spacers and usually clustering with *cas* (CRISPR-associated) genes [29]. No matching sequence between the Mae-Yong924-1 genome and viral spacer sequences was found within the CRISPRs database.

RAST, BLASTp, Hmmer and HHpred analyses proposed that Mae-Yong924-1 has 59 predicted ORFs, which encode hypothetical proteins/peptides of 37-1011 residues in length. Sequence comparison indicated that, of the 59 predicted ORFs, only 42 (71%) had homologs in current databases, and only 12 (20%) could be functionally annotated. The rest of the ORFs shared no sequence similarity to any previously identified proteins. The 59 predicted ORFs could be divided into 5 functional groups: lysis (1 ORF), packaging (2 ORFs), structure (2 ORFs), regulation and replication (7 ORFs) and uncharacterized (47 ORFs) (Figure 3).

Of the 12 annotated ORFs of the *Microcystis* cyanophage Mae-Yong924-1, 7 were predicted to encode putative proteins involved in regulation and replication, including a regulatory protein (ORF 3), a protein containing nucleotide modification associated domain 5 (ORF 5), DNA cytosine methyltransferase (ORF 10), DNA primase-helicase (ORF 14), a DNA repair protein (ORF 25), polyphosphate kinase (OFR 44) and protease 4 (ORF 37). Two ORFs of the *Microcystis* cyanophage Mae-Yong924-1 were predicted to encode the putative terminase small subunit (ORF 45) and terminase large subunit (ORF 47) involved in the packaging of double-stranded DNA into viral procapsids [30].

Surprisingly, only two ORFs in the Mae-Yong924-1 genome were predicted to encode putative structural proteins including a phage-related minor tail protein (ORF 54) and a major capsid protein, gpN (ORF 38). No other gene sharing significant homology with viral structural proteins was found, which is very different from most known cyanophages. This character echoes the unique morphological structure feature of Mae-Yong924-1, significantly different from that of all the reported phages (Figure 1). Correspondingly, the Mae-Yong924-1 genome had extremely low similarity to sequences in current databases.

As described in the previous section, both BLASTn and BLASTx scanning of the Mae-Yong924-1 genome resulted in “No significant similarity found”, which demonstrated the extremely low similarity between Mae-Yong924-1 and other phages in current databases. To further estimate the nucleotide sequence similarity between Mae-Yong924-1 and other phages in current (10 August 2021) public databases, the Pairwise Sequence Comparison (PASC) classification tool [31] (https://www.ncbi.nlm.nih.gov/sutils/pasc/viridty.cgi; 25 August 2021) was used. Mae-Yong924-1 shared the highest nucleotide sequence similarity, as low as 11.94%, with the most closely related phage in the PASC search. This value is well below the >50% boundaries to define a genus. The PASC value indicates that Mae-Yong924-1 belongs to an unknown new genus. The complete genome sequences of 10 cyanophages including 5 *Microcystis* cyanophages, 43 representative bacteriophages of the 14 families and the genus *Lilyvirus* of *Caudovirales* (which are the latest classification by the ICTV) and 3 *Alloherpesviridae* viruses of *Herpesvirales* were obtained from the NCBI database. A proteomic tree was constructed based on the complete genome sequences. In the phylogenetic tree, Mae-Yong924-1 formed a root node alone and monopolized a root branch. The phylogenetic tree illustrated that there was a far evolutionary distance between Mae-Yong924-1 and the other phages of *Caudovirales* (Figure 4). Mae-Yong924-1 may reveal a novel clade (at least a novel family). Of the 11 cyanophages in the proteomic tree, only 2, i.e., MaMV-DC and Ma-LMM01, have a definite taxonomic status designated by the ICTV, and the other 9 strains are unclassified. MaMV-DC and Ma-LMM01 were classified as *Myoviridae* family members in the current ICTV classification. However, in the proteomic tree of this study, MaMV-DC and Ma-LMM01 were found to be very distinct from other *Myoviridae* phages. The proteomic tree reveals global genomic similarity relationships among viruses. Phage proteomic trees are advocated by the ICTV for use as the basis of a genome-based taxonomical system for phages. We propose that the creation of a new family should comprise MaMV-DC and Ma-LMM01 in the next versions of taxonomic updates.

In summary, based on the morphology and sequence characteristics, we propose that the *Microcystis* cyanophage Mae-Yong924-1 may represent a new clade (at least a novel family) of phages. The isolation of Mae-Yong924-1 is an exciting discovery due to its uniqueness and capability of cross-taxonomic order infectivity.

## Figures and Tables

**Figure 1 viruses-14-00283-f001:**
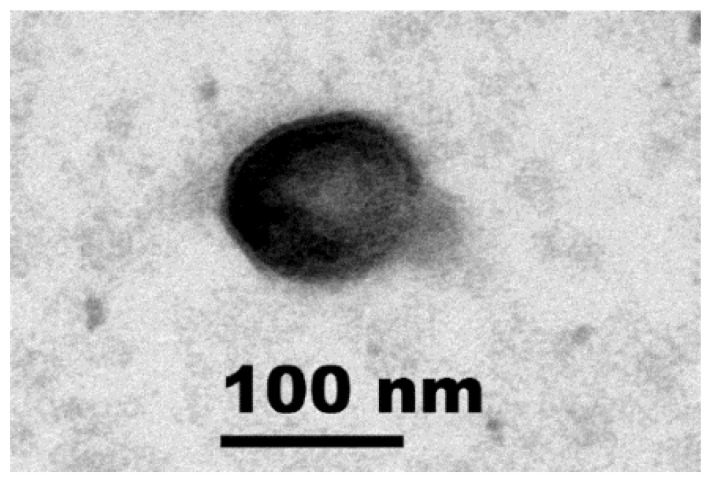
Morphology of cyanophage Mae-Yong924-1. Transmission electron micrograph of negatively stained phage particle. Scale bar, 100 nm.

**Figure 2 viruses-14-00283-f002:**
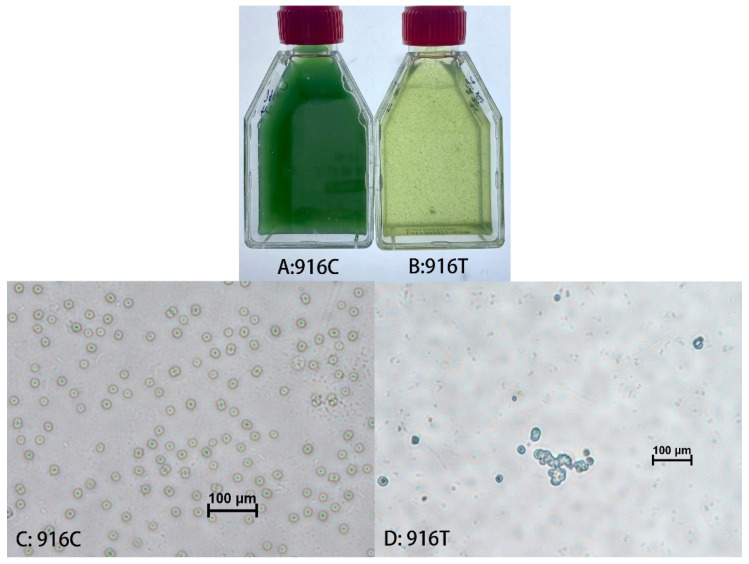
Macro- and micrographs of *M. elabens* FACHB-916 cultures. (**A**) Macrograph of normal FACHB-916 cultures; (**B**) Macrograph of FACHB-916 co-inoculated with cyanophage Mae-Yong924-1; (**C**) Micrograph of normal FACHB-916 cultures; (**D**) Micrograph of FACHB-916 co-inoculated with cyanophage Mae-Yong924-1.

**Figure 3 viruses-14-00283-f003:**
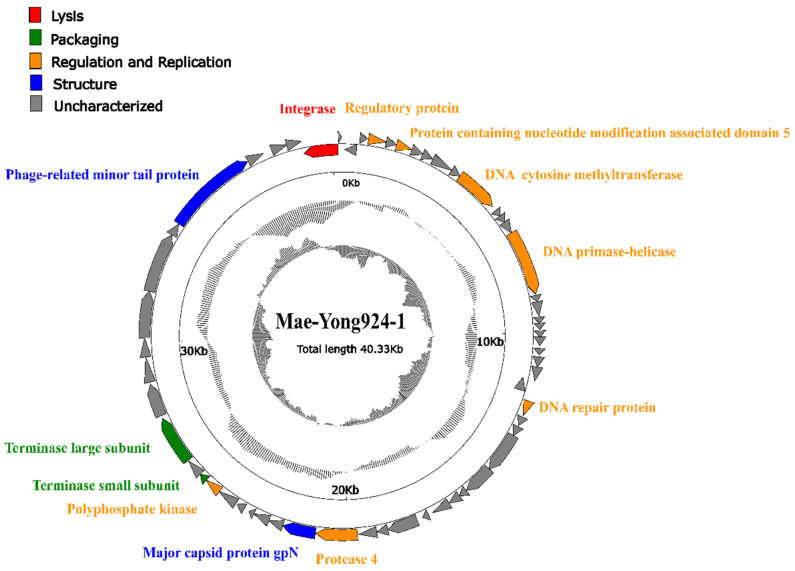
Genome map of cyanophage Mae-Yong924-1. The outermost circle represents the 59 ORFs encoded in the genome, with different colors representing different functions (clockwise arrow indicates the forward reading frame, and counterclockwise arrow indicates the reverse reading frame); the dark circles in the middle represent the GC content (outwards indicates greater than the average GC content compared with the whole genome, and inwards indicates the opposite); the innermost circle represents the GC skew (G − C/G + C; outwards indicates >0, and inwards indicates <0).

**Figure 4 viruses-14-00283-f004:**
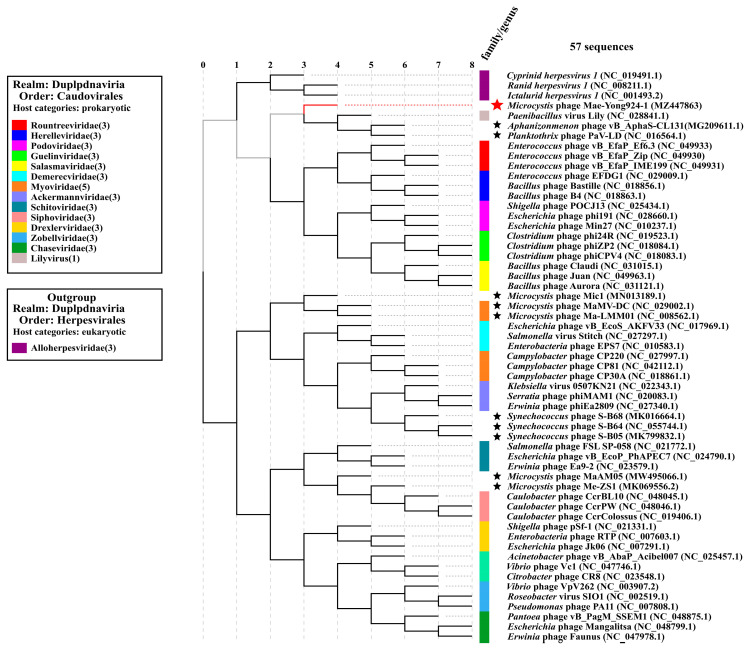
Proteomic tree based on the complete genome sequences of Mae-Yong924-1 (red star), 10 other cyanophages (black star) including 5 *Microcystis* cyanophages, 43 representative bacteriophages of *Caudovirales* and 3 viruses of *Herpesvirales*.

**Table 1 viruses-14-00283-t001:** Results of host range test of Mae-Yong924-1 against 14 cyanobacterial strains.

Orders	Families	Species	Strains	Susceptible	Origin
*Chroococcales*	*Microcystaceae*	*Microcystis aeruginosa*	FACHB-924	+	Australia
			FACHB-469	−	France
		*M. elabens*	FACHB-916	+	Japan
		*Microcystis* sp.	PCC-7806	+	France
		*M* *. ichthyoblabe*	FACHB-1409	−	China
*Nostocales*	*Nostocaceae*	*Nostoc* sp.	FACHB-596	+	China
		*Dolichospermum flos-aquae*	FACHB-1255	−	China
		*Aphanizomenon flos-aquae*	FACHB-1209	+	China
			FACHB-1040	−	China
*Oscillatoriales*	*Microcoleaceae*	*Planktothrix agardhii*	FACHB-920	−	Japan
		*Planktothricoides raciborskii*	FACHB-881	+	China
*Synechococcales*	*Synechococcaceae*	*Synechococcus* sp.	FACHB-805	−	Australia
*Hormogonales*	*Scytonemataceae*	*Plectonema boryanum*	FACHB-240	−	America
			FACHB-402	−	America

(+) indicates infection, (−) indicates non-infection.

## Data Availability

Not applicable.

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
