# Peer review of "A Novel Freshwater Cyanophage, Mae-Yong924-1, Reveals a New Family"

_viruses, 2022, doi:10.3390/v14020283_

Round 1

Reviewer 1 Report

The manuscript of Qian M. et al. describes the isolation and charterization of a novel cyanobacterial phage which represent a new class of bacteriophages. The methodology and implementation were appropriate and the standard of communication is also good. The publication is worth to the attention of the scientific community.

Special comments

Line 65: represent

Line 87: exponential phase is better (however, it used to be not properly use in many places)

Table 1: Families

Figure 4: A rooted tree with an outgroup would be more appropriate.

Reviewer 2 Report

This is an interesting manuscript describing the isolation and characterization of a new cyanobacteria-infecting bacteriophage with potential as biocontrol agent for cyanobacterial blooms. Most significantly the phage’s genome appears to not have any resemblance with other known Caudovirales phages and therefore might represent a new taxonomic clade/family. The results are overall presented in a clear manner and the conclusions are valid. I have only a few remarks; after those are answered and the manuscript changed accordingly I recommend publication.

Line 12: “specifically infecting”…

Line 18: taxon names need to be italicized

Line 22: “were predicted functions” – check grammar

Line 24: “may reveal”

There are numerous grammatical mistakes throughout the remainder of the manuscript. Please have it proofread by a native speaker. Please also make sure to italicize taxonomic names throughout the manuscript.

Line 31: “edible risks” – please rephrase

Line 105: “12th day”?

Figure 1: I don’t see the “broomstick” tail on this picture. Is there one where this feature is better visible?

To estimate the suitability of the phage as a biocontrol agent, it would be interesting to see if the bacteria become phage-resistant over time. Have you performed any such experiment?
